# SCALABLE COMPUTATION OF MONGE MAPS WITH GENERAL COSTS

**Jiaojiao Fan**[*]**, Shu Liu**[*]**, Shaojun Ma**[*]**, Yongxin Chen, Haomin Zhou**
Georgia Institute of Technology
`{jiaojiaofan, sliu459, shaojunma, yongchen}@gatech.edu`
`hmzhou@math.gatech.edu`

## ABSTRACT

Monge map refers to the optimal transport map between two probability distributions and provides a principled approach to transform one distribution to another. In spite of the rapid developments of the numerical methods for optimal transport problems, computing the Monge maps remains challenging, especially for high dimensional problems. In this paper, we present a scalable algorithm for computing the Monge map between two probability distributions. Our algorithm is based on a weak form of the optimal transport problem, thus it only requires samples from the marginals instead of their analytic expressions, and can accommodate optimal transport between two distributions with different dimensions. Our algorithm is suitable for general cost functions, compared with other existing methods for estimating Monge maps using samples, which are usually for quadratic costs. The performance of our algorithms is demonstrated through a series of experiments with both synthetic and realistic data.

## 1 INTRODUCTION

In recent years we have witnessed great success of optimal transport (OT) (Villani, 2008) based applications in machine learning community (Arjovsky et al., 2017; Krishnan & Martínez, 2018; Li et al., 2019; Makkuva et al., 2020; Inoue et al., 2020; Ma et al., 2020; Fan et al., 2020; Haasler et al., 2020). As a crucial concept of OT, Wasserstein distance is used to evaluate the discrepancy between distributions due to its good properties such as symmetry and robustness. In this work, given any two probability distributions $\rho_a$ and $\rho_b$ defined on $\mathbb{R}^n$ and $\mathbb{R}^m$, we consider the Monge problem

$$C_{\text{Monge}}(\rho_a, \rho_b) \triangleq \min_{T:\mathbb{R}^n \to \mathbb{R}^m, T_\sharp \rho_a = \rho_b} \int_{\mathbb{R}^n} c(x, T(x))\rho_a(x)\, dx. \tag{1}$$

Here $c(x, y)$ denotes the cost of transporting from $x$ to $y$ and $T$ is the transport map. We define the pushforward of distribution $\rho_a$ by $T$ as $T_\sharp \rho_a(E) = \rho_a(T^{-1}(E))$ for any measurable set $E \subset \mathbb{R}^m$. The Monge problem seeks for the cost-minimizing transport plan $T_*$ from $\rho_a$ to $\rho_b$. The optimal $T_*$ is also known as **Monge map** of equation 1.

Solving equation 1 in high dimensional space yields a challenging problem due to the curse of dimensionality for discretization. A modern formulation of Monge problem equation 1 as a linear programming problem known as Kantorovich problem (Villani, 2003), and adding an entropic regularization, one is capable of computing the problem via iterative Sinkhorn algorithm (Cuturi, 2013). Such type of treatment has been widely accepted since it is friendly to high dimensional cases (Altschuler et al., 2017; Genevay et al., 2018; Li et al., 2019; Xie et al., 2020), but the algorithm does not scale well to a large number of samples and is not suitable to handle continuous probability measures (Genevay et al., 2016). Recently, the dual form of Kantorovich problem is found to facilitate the computation of OT problem. Meanwhile, with the rising popularity of neural networks, many regularization-based OT problems have been formulated, such as entropic regularized OT (Seguy et al., 2017), Laplacian regularization (Flamary et al., 2014), Group-Lasso regularized OT (Courty et al., 2016), Tsallis regularized OT (Muzellec et al., 2017) and OT with $L^2$ regularization (Dessein et al., 2018).

---

[*]Equal contribution

In this work, we propose a computationally efficient and scalable algorithm for estimating the Wasserstein distance as well as the optimal map in continuous spaces without introducing any regularization terms. Particularly, we apply the Lagrangian multiplier directly to Monge problem, and obtain a minimax problem. Our contribution is summarized as follows: 1) We develop a scalable algorithm to compute the optimal transport map associated with general transport costs between any two distributions given their samples; 2) Our method is capable of computing OT problems between distributions over spaces that do not share the same dimension. 3) We provide a rigorous error analysis of the algorithm based on duality gaps; 4) We demonstrate its performance and its scaling properties in truly high dimensional setting through experiments.

## 2 PROPOSED METHOD

In order to formulate a tractable algorithm for the general Monge problem equation 1, we first notice that equation 1 is a constrained optimization problem. Thus, it is natural to introduce the Lagrange multiplier $f$ for the constraint $T_\sharp \rho_a = \rho_b$ and then reformulate equation 1 as a saddle point problem

$$\sup_f \inf_T \mathcal{L}(T, f), \tag{2}$$

with $\mathcal{L}$ defined as

$$\mathcal{L}(T, f) = \int_{\mathbb{R}^n} c(x, T(x))\rho_a(x)dx + \int_{\mathbb{R}^m} f(y)(\rho_b - T_\sharp \rho_a) \, dy$$

$$= \int_{\mathbb{R}^n} [c(x, T(x)) - f(T(x))] \rho_a(x) \, dx + \int_{\mathbb{R}^m} f(y)\rho_b(y) \, dy. \tag{3}$$

The following theorem ensures the consistency of the max-min formulation equation 2.

**Theorem 1** (Consistency). *Assume that the optimal solution to the Monge problem equation 1 exists. Suppose the saddle point solution to equation 2 is $(T_*, f_*)$, then $T_*$ is the Monge map to the problem equation 1 and $f_* = \phi_*$, where $\phi_*$ is the optimal solution $\phi$ to the Kantorovich dual problem (see (5.3) in Villani (2008)).*

The proof of this theorem can be found in appendix A. In implementation, we parametrize both the map $T$ and the dual variable $f$ by the neural networks $T_\theta, f_\eta$, with $\theta, \eta$ being the parameters of the networks. We aim at solving the following saddle point problem

$$\max_\eta \min_\theta \mathcal{L}(T_\theta, f_\eta) := \frac{1}{N} \sum_{k=1}^N c(X_k, T_\theta(X_k)) - f_\eta(T_\theta(X_k)) + f_\eta(Y_k) \tag{4}$$

where $N$ is size of the datasets and $\{X_k\}, \{Y_k\}$ are samples drawn by $\rho_a$ and $\rho_b$ separately. The algorithm is summarized in Algorithm 1. The computational complexity of our algorithm is similar with GAN-type methods.

---

**Algorithm 1** Computing Wasserstein distance and optimal map from $\rho_a$ to $\rho_b$

---

1: **Input**: Marginal distributions $\rho_a$ and $\rho_b$, Batch size $B$, Cost function $c(x, T(x))$.
2: Initialize $T_\theta, f_\eta$.
3: **for** $K$ steps **do**
4:     Sample $\{X_k\}_{k=1}^B$ from $\rho_a$. Sample $\{Y_k\}_{k=1}^B$ from $\rho_b$.
5:     Update (via gradient descent) $\theta$ to decrease (4) for $K_1$ steps.
6:     Update (via gradient ascent) $\eta$ to increase (4) for $K_2$ steps.
7: **end for**

---

**Remark 1** (Relation with WGAN). *Although the proposed saddle scheme equation 2 shares similarity with the Wasserstein GAN (Arjovsky et al., 2017), both the designing purpose and the mathematical logic behind both methods are distinct. Detailed comparisons are discussed in appendix B.*

## 3 ERROR ANALYSIS VIA DUALITY GAPS

In this section, we assume $m = n = d$, i.e. we consider Monge problem in the same space $\mathbb{R}^d$. Suppose we solve equation 2 to a certain stage and obtain the pair $(T, f)$, inspired by Hütter &

Rigollet (2020) and Makkuva et al. (2020), we provide an *a posterior* estimate to a weighted $L^2$ error between our computed map $T$ and the optimal Monge map $T_*$. Before we present our result, we should introduce our assumptions.

**Assumption 1.** *(1) We assume $c \in C^2(\mathbb{R}^d \times \mathbb{R}^d)$, i.e., $c$ is second order continuously differentiable;*

*(2) For any $x, y \in \mathbb{R}^d$, $\partial_x c(x, \cdot)$ is injective map; $\partial_{xy} c(x, y)$, as a $d \times d$ matrix, is invertible and symmetric, and we denote $\sigma(x, y) = \sigma_{\min}(\partial_{xy} c(x, y)) > 0$ as the minimum singular value of $\partial_{xy} c(x, y)$; $\partial_{yy} c(x, y)$ is independent of $x$.*

*(3) $\rho_a, \rho_b$ are compactly supported on $\mathbb{R}^d$, and $\rho_a$ admits density.*

*(4) Assume our dual variable $f \in C^2(\mathbb{R}^d)$ is always taken from c-concave functions (c.f. Definition 5.7 of Villani (2008)), i.e., there exists $\varphi \in C^2(\mathbb{R}^d)$ such that $f(\cdot) = \inf_x \{\varphi(x) + c(x, \cdot)\}$.*

*(5) For any $y \in \mathbb{R}^d$ there is unique minimizer $\hat{x}_y \in argmin\{\varphi(x) + c(x, y)\}$. And we further assume $\varphi(\cdot) + c(\cdot, y)$ is strictly convex and its Hessian can be upper bounded by $\lambda(\cdot) > 0$, i.e.,*

$$\lambda(y)I \succeq \nabla_{xx}^2(\varphi(x) + c(x, y))|_{x=\hat{x}_y} \succ O.$$

We denote the duality gaps:

$$\mathcal{E}_1(T, f) = \mathcal{L}(T, f) - \inf_{\widetilde{T}} \mathcal{L}(\widetilde{T}, f), \quad \mathcal{E}_2(f) = \sup_{\widetilde{f}} \inf_{\widetilde{T}} \mathcal{L}(\widetilde{T}, \widetilde{f}) - \inf_{\widetilde{T}} \mathcal{L}(\widetilde{T}, f).$$

Let $T_*$ as the Monge map of the OT problem equation 1. We have the following theorem:

**Theorem 2** (Posterior Error Analysis via Duality Gaps). *Under Assumption 1, there exists a strict positive weight function $\beta(x) > \min_y \left\{ \frac{\sigma(x,y)}{2\lambda(y)} \right\}$ such that the weighted $L^2$ error between computed map $T$ and optimal map $T_*$ is upper bounded by*

$$\|T - T_*\|_{L^2(\beta \rho_a)} \leq \sqrt{2(\mathcal{E}_1(T, f) + \mathcal{E}_2(f))}.$$

*Here $\sigma$ and $\lambda$ are defined in Assumption 1. Exact formulation of $\beta$ is provided in equation 23 in the appendix C.*

The proof of this theorem can be found in the appendix C.

**Remark 2.** *We can verify that $c(x, y) = \frac{1}{2}\|x - y\|^2$ or $c(x, y) = -x \cdot y$ satisfy the conditions mentioned above. Then Theorem 2 recovers similar results proved in Hütter & Rigollet (2020) and Makkuva et al. (2020).*

## 4 EXPERIMENTS

Recently, several works have illustrated the scalability of our dual formula with different realizations of the transportation costs. With a focus on the quadratic cost, Rout et al. (2022) obtains comparable performance in image generative models, which asserts the efficacy in unequal dimension tasks. Similarly, Korotin et al. (2022) apply the formula with quadratic cost in multiple domain adaptation tasks and achieve the respectable effect. Concurrently, Gazdieva et al. (2022) utilizes the dual formula with more diverse costs in image super-resolution task. Our dual formula can be viewed as the extension to all the above approaches.

In this section, we show the effectiveness of our method on the inpainting task with random rectangle masks. We take the distribution of occluded images to be $\rho_a$ and the distribution of the full images to be $\rho_b$. In many inpainting works, it's assumed that an unlimited amount of paired training data is accessible (Zeng et al., 2021). However, most real-world applications do not involve the paired datasets. Accordingly, we consider the **unpaired** inpainting task, i.e. no pair of masked image and original image is accessible. The training and test data are generated according to Rout et al. (2022, Section 5.2). We choose cost function to be mean squared error (MSE) in the unmasked area

$$c(x, y) = \alpha \cdot \frac{\|x \odot M - y \odot M\|_2^2}{n},$$

where $M$ is a binary mask with the same size as the image. $M$ takes the value 1 in the unoccluded region, and 0 in the unknown/missing region. $\odot$ represents the point-wise multiplication, $\alpha$ is a

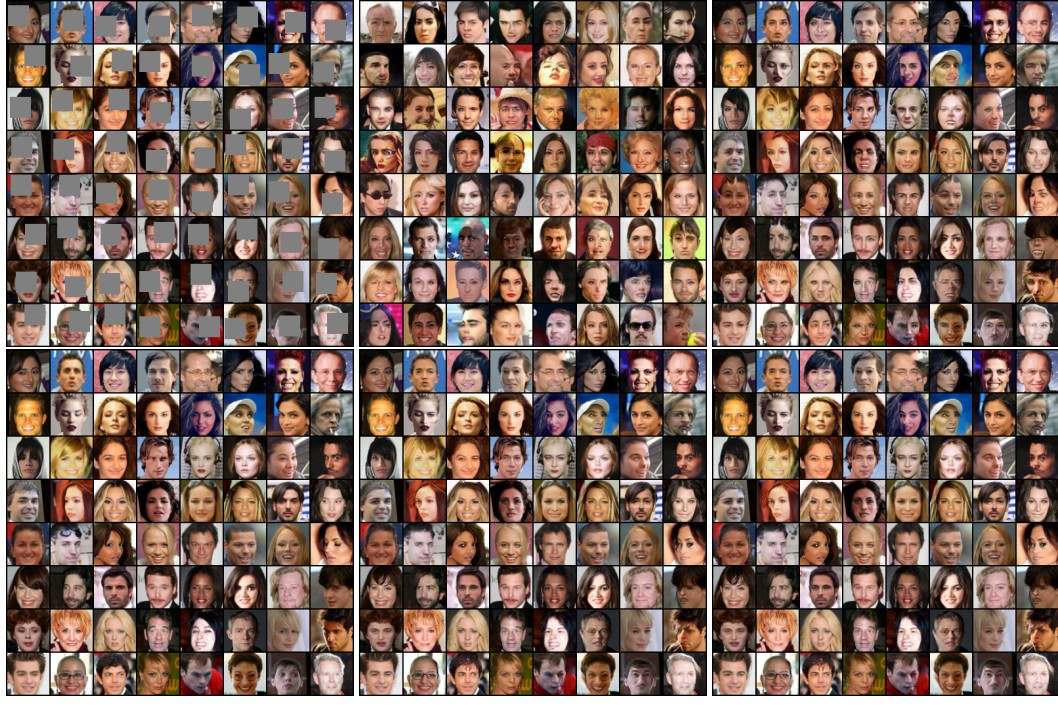

| (a) Degraded and original images | (b) Composite images $G(x)$ | (c) Pushforward images $T(x)$ |

Figure 1: Unpaired image inpainting on **test** dataset of CelebA $64 \times 64$. In panel (b) and (c), we show the results with $\alpha = 10$ in the first row and $\alpha = 10000$ in the second row. A small transportation cost would result that pushforward map neglects the connection to the unmasked area, which is illustrated by a clear mask border in pushforward images.

tunable coefficient, and $n$ is dimension of $x$. Intuitively, this works as a regularization that the pushforward images should be consistent with input images in the unmasked area. Empirically, the map learnt with a larger $\alpha$ can generate more realistic images with natural transition in the mask border and exhibit more details on the face.

We conduct the experiments on CelebA $64 \times 64$ and $128 \times 128$ datasets (Liu et al., 2015). The input images ($\rho_a$) are occluded by randomly positioned square masks. Each of the source $\rho_a$ and target $\rho_b$ distributions contains $80k$ images. We present the empirical results of inpainting in Figure 1 and 2. Denote $M^C$ as the complement of $M$, i.e. $M^C = 1$ in the occluded area and 0 otherwise. We take the composite image $G(x) = T(x) \odot M^C + x \odot M$ as the output image. Additionally, we provide the pushforward images $T(x)$ to illustrate the regularization effect of transportation cost in Figure 1. We also provide quantitative results in Section D.1.

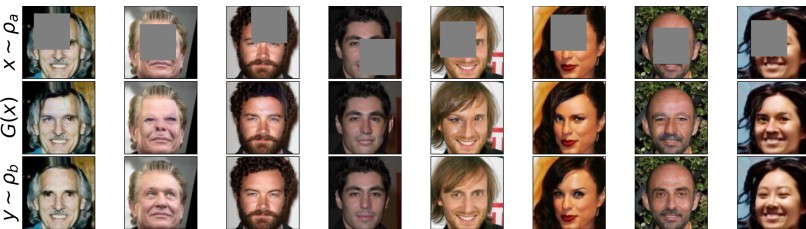

Figure 2: Unpaired image inpainting on **test** dataset of CelebA $128 \times 128$.

## 5 CONCLUSION

In this paper we present a novel method to compute Monge map between two given distributions with freely chosen cost functions. In particular, we consider applying Lagrange multipliers on the Monge problem, which leads to a max-min saddle point problem. By further introducing neural networks into our optimization, we obtain a scalable algorithm that can handle most general costs and even the

case where the dimensions of marginals are unequal. Our scheme is shown to be effective through a series of experiments with both low dimensional and high dimensional settings. It will become an useful tool for machine learning applications such as domain adaption that requires transforming data distributions. It will also be potentially used in areas outside machine learning, such as robotics.

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

# A    PROOF OF THEOREM 1

We first introduce the Kantorovich problem of optimal tranport:

$$C(\rho_a, \rho_b) = \min_{\pi \in \Pi(\rho_a, \rho_b)} \left\{ \iint_{\mathbb{R}^n \times \mathbb{R}^m} c(x, y) d\pi(x, y) \right\}. \tag{5}$$

Here we denote $\Pi(\rho_a, \rho_b)$ as the set of all probability distributions on $\mathbb{R}^n \times \mathbb{R}^m$ with fixed marginals equal to $\rho_a$ and $\rho_b$. One can verify that if the optimal solution $T_*$ of the Monge problem exists, then equation 5 is related to equation 1 in the sense of $(\mathrm{Id}, T_*)_\sharp \rho_a = \pi_*$ where $\pi_*$ is the optimal solution to equation 5, and $C(\rho_a, \rho_b) = C_{\mathrm{Monge}}(\rho_a, \rho_b)$.

**Theorem 1** (Consistency). *We assume that the optimal solution to the Monge problem equation 1 exists. Suppose the optimal solution to equation 2 is $(T_*, f_*)$, then $T_*$ is the Monge map to the problem equation 1 and $f_* = \phi_*$, where $\phi_*$ is the optimal solution to the **dual Kantorovich problem***

$$\sup_{\phi \in L^1(\rho_b)} \left\{ \int_{\mathbb{R}^m} \phi(y) \rho_b(y) \, dy - \int_{\mathbb{R}^n} \phi^{c,-}(x) \rho_a(x) \, dx \right\}, \tag{6}$$

*and we define $\phi^{c,-}$ via supremum convolution: $\phi^{c,-}(x) = \sup_y (\phi(y) - c(x, y))$.*

*Proof of Theorem 1.* According to following equation

$$\inf_T \mathcal{L}(T, f) = -\int_{\mathbb{R}^n} \sup_\xi \{ f(\xi) - c(x, \xi) \} \rho_a(x) dx + \int_{\mathbb{R}^m} f(y) \rho_b(y) dy$$

$$= \int_{\mathbb{R}^m} f(y) \rho_b(y) dy - \int_{\mathbb{R}^n} f^{c,-}(x) \rho_a(x) dx, \tag{7}$$

we are able to tell that the optimal solution $f_*$ equals $\phi_*$. Furthermore, at the optimal point $(T_*, f_*)$, we have

$$T_{*\sharp} \rho_a = \rho_b, \quad T_*(x) \in \mathrm{argmax}_{\xi \in \mathbb{R}^m} \{ f_*(\xi) - c(x, \xi) \}, \; \rho_a \text{ almost surely.}$$

The second equation leads to

$$f_*^{c,-}(x) = f_*(T_*(x)) - c(x, T_*(x)), \; \rho_a \text{ almost surely.}$$

Then we have

$$\int_{\mathbb{R}^n} c(x, T_*(x)) \rho_a(x) \, dx = \int_{\mathbb{R}^n} f_*(T_*(x)) \rho_a(x) \, dx - \int_{\mathbb{R}^n} f_*^{c,-}(x) \rho_a(x) \, dx$$

$$= \int_{\mathbb{R}^m} f_*(y) \rho_b(y) \, dy - \int_{\mathbb{R}^n} f_*^{c,-}(x) \rho_a(x) \, dx$$

$$\leq \int_{\mathbb{R}^n \times \mathbb{R}^m} [f_*(y) - f_*^{c,-}(x)] d\pi(x, y) \leq \int_{\mathbb{R}^n \times \mathbb{R}^m} c(x, y) d\pi(x, y)$$

for any $\pi \in \Pi(\rho_a, \rho_b)$. Here the second equality is due to $T_{*\sharp} \rho_a = \rho_b$, the last inequality is due to the definition of $f_*^{c,-}(x) = \sup_y \{ f_*(y) - c(x, y) \}$.

We now take the infimum value of $\int_{\mathbb{R}^n \times \mathbb{R}^m} c d\pi$ and we obtain

$$\int_{\mathbb{R}^n} c(x, T_*(x)) \rho_a(x) \, dx \leq C(\rho_a, \rho_b),$$

Now since we assume the optimal solution to equation 1 exits, thus $C(\rho_a, \rho_b) = C_{\mathrm{Monge}}(\rho_a, \rho_b)$. Thus we deduce that

$$\int_{\mathbb{R}^n} c(x, T_*(x)) \rho_a(x) \, dx = C_{\mathrm{Monge}}(\rho_a, \rho_b).$$

As a result, $T_*$ solves equation 1 and thus is the Monge map. $\qquad \square$

# B  RELATION BETWEEN OUR METHOD AND GENERATIVE ADVERSARIAL NETWORKS

It is worth pointing out that our scheme and Wasserstein Generative Adversarial Networks (WGAN) Arjovsky et al. (2017) are similar in the sense that they are both doing minimization over the generator/map and maximization over the discriminator/dual potential. However, there are two main distinctions between them. Such differences are not reflected from the superficial aspects such as the choice of reference distributions $\rho_a$, but come from the fundamental logic hidden behind the algorithms.

- We want to first emphasize that the mechanisms of two algorithms are different: Typical Wasserstein GANs (WGAN) are usually formulated as

$$\min_G \underbrace{\max_{\|D\|_{\text{Lip}\leq 1}} \int D(y)\rho_b(y)dy - \int D(G(x))\rho_a(x)dx}_{1-\text{Wasserstein distance } W_1(G_\sharp\rho_a, \rho_b)} \tag{8}$$

and ours reads

$$\max_f \min_T \underbrace{\int f(y)\rho_b(y)dy - \int f(T(x))\rho_a(x)dx + \int c(X, T(x))\rho_a(x)dx}_{\text{general Wasserstein distance } C(\rho_a, \rho_b)} \tag{9}$$

  The inner maximization of equation 8 computes $W_1$ distance via Kantorovich duality and the outer loop minimize the $W_1$ gap between desired $\rho_b$ and $G_\sharp\rho_a$; However, the logic behind our scheme equation 9 is different: the inner optimization computes for the $c-$transform of $f$, i.e. $f^{c,-}(x) = \sup_\xi(f(\xi) - c(x, \xi))$; And the outer maximization computes for the Kantorovich dual problem $C(\rho_a, \rho_b) = \sup_f \{\int f(y)\rho_b(y)dy - \int f^{c,-}(x)\rho_a(x)dx\}$.

  Even under $W_1$ circumstance, one can verify the intrinsic difference between two proposed methods: when setting the cost $c(x, y) = \|x - y\|$, and $\rho_a = G_\sharp\rho_a$ in equation 9, the entire "max-min" optimization of equation 9 (underbraced part) is equivalent to the inner maximization problem of equation 8 (underbraced part), but not for the entire saddle point scheme.

  It is also important to note that WGAN aims to minimize the distance between generated distribution and the target distribution and the ideal value for equation 8 is $0$. On the other hand, one of our goal is to estimate the optimal transport distance between the initial distribution $\rho_a$ and the target distribution $\rho_b$. Thus the ideal value for equation 9 should be $C(\rho_a, \rho_b)$, which is not $0$ in most of the cases.

- We then argue about the optimality of the computed map $G$ and $T$: In equation 8, one is trying to obtain a map $G$ by minimizing $W_1(\rho_b, G_\sharp\rho_a)$ w.r.t. $G$, and hopefully, $G_\sharp\rho_a$ can approximate $\rho_b$ well. However, there isn't any restriction exerted on $G$, thus one can not expect the computed $G$ to be the optimal transport map between $\rho_a$ and $\rho_b$; On the other hand, in equation 9, we not only compute $T$ such that $T_\sharp\rho_a$ approximates $\rho_b$, but also compute for the optimal $T$ that minimizes the transport cost $\mathbb{E}_{\rho_a}[c(X, T(X))]$. In equation 9, the computation of $T$ is naturally incorporated in the max-min scheme and there exists theoretical result (recall Theorem 1 in the paper,) that guarantees $T$ to be the optimal transport map.

In summary, even though the formulation of both algorithms are similar, the designing logic (minimizing distance vs computing distance itself) and the purposes (computing arbitary pushforward map vs computing the optimal map) of the two methods are distinct. Thus the theoretical and empirical study of GANs cannot be trivially translated to proposed method. In addition to the above discussions, we should also refer the readers to Gazdieva et al. (2022), in which a comparison between a similar saddle point method and the regularized GANs are made in section 6.2 and summarized in Table 1.

# C  PROOF OF THEOREM 2

**Theorem 2** (Posterior Error Analysis via Duality Gaps). *Suppose $c(\cdot, \cdot), \rho_a, \rho_b$ satisfy the conditions mentioned in Section 3. We assume our dual variable $f \in C^2(\mathbb{R}^d)$ and is always taken from c-*

*concave functions (c.f. Definition 5.7 of Villani (2008)), to be more specific, we suppose there exists $\varphi \in C^2(\mathbb{R}^d)$ such that $f(\cdot) = \inf_x \{\varphi(x) + c(x, \cdot)\}$ We further denote*

$$\sigma(x, y) = \sigma_{min}(\partial_{xy} c(x, y)) \tag{10}$$

*as the minimum singular value of matrix $\partial_{xy} c(x, y)$, since the matrix is invertible, $\sigma(x, y) > 0$ for any $x, y \in \mathbb{R}^d$.*

*Consider for any $y \in \mathbb{R}^d$ there is unique minimizer $\hat{x}_y \in argmin\{\varphi(x) + c(x, y)\}$, we further assume $\varphi(\cdot) + c(\cdot, y)$ is strictly convex and its Hessian can be upper bounded by $\lambda(\cdot) > 0$, i.e.,*

$$\lambda(y) I_n \succeq \nabla^2_{xx}(\varphi(x) + c(x, y))\Big|_{x=\hat{x}_y} \succ O_n \tag{11}$$

*If we denote the duality gaps*

$$\mathcal{E}_1(T, f) = \mathcal{L}(T, f) - \inf_{\widetilde{T}} \mathcal{L}(\widetilde{T}, f)$$

$$\mathcal{E}_2(f) = \sup_{\widetilde{f}} \inf_{\widetilde{T}} \mathcal{L}(\widetilde{T}, \widetilde{f}) - \inf_{\widetilde{T}} \mathcal{L}(\widetilde{T}, f)$$

*Denote $T_*$ as the optimal Monge map of the OT problem equation 1. Then there exists a strict positive weight function $\beta(\cdot) > 0$ ( depending on $c, T_*, f$ and $\varphi$, such that the weighted $L^2$ error between computed map $T$ and optimal map $T_*$ is upper bounded by*

$$\|T - T_*\|_{L^2(\beta \rho_a)} \leq \sqrt{2(\mathcal{E}_1(T, f) + \mathcal{E}_2(f))}.$$

**Lemma 1.** *Suppose $n \times n$ matrix $A$ is self-adjoint, i.e. $A = A^T$, with minimum singular value $\sigma_{\min}(A) > 0$. Also assume $n \times n$ matrix $H$ is self-adjoint and satisfies $\lambda I_n \succeq H \succ O_n$. Then $A H^{-1} A \succeq \frac{\sigma_{\min}(A)^2}{\lambda} I_n$.*

*Proof of Lemma 1 .* One can first verify that $H^{-1} \succeq \frac{1}{\lambda} I_n$ by digonalizing $H^{-1}$. To prove this lemma, we only need to verify that for arbitrary $v \in \mathbb{R}^n$,

$$v^{\mathsf{T}} A H^{-1} A v = (Av)^{\mathsf{T}} H^{-1} A v \geq \frac{|Av|^2}{\lambda} \geq \frac{\sigma_{\min}(A)^2}{\lambda} |v|^2$$

Thus $A H^{-1} A - \frac{\sigma_{\min}(A)^2}{\lambda} I_n$ is non-negative definite. $\qquad \square$

The following lemma is crucial for proving our results, it analyzes the concavity of the target function $f(\cdot) - c(\cdot, y)$ with $f$ $c$-concave.

**Lemma 2** (Concavity of $f(\cdot) - c(x, \cdot)$ as $f$ $c$-concave). *Suppose the cost function $c(x, y)$ and $f$ satisfy the conditions mentioned in Theorem 2. Denote the function $\Psi_x(y) = f(y) - c(x, y)$, keep all notations defined in Theorem 2, then we have*

$$\nabla^2 \Psi_x(y) \preceq -\frac{\sigma(x, y)^2}{\lambda(y)} I_n.$$

*Proof of Lemma 2.* First, we notice that $f$ is $c$-convex, thus, there exists $\varphi$ such that $f(y) = \inf_x \{\varphi(x) + c(x, y)\}$. Let us also denote $\Phi(x, y) = \varphi(x) + c(x, y)$.

Now for a fixed $y \in \mathbb{R}^n$, We pick one

$$\hat{x}_y \in \operatorname{argmin}_x \{\varphi(x) + c(x, y)\}$$

Since we assumed that $\varphi \in C^2(\mathbb{R}^n)$ and $c \in C^2(\mathbb{R}^n \times \mathbb{R}^n)$, we have

$$\partial_x \Phi(\hat{x}_y, y) = \nabla \varphi(\hat{x}_y) + \partial_x c(\hat{x}_y, y) = 0 \tag{12}$$

At the same time, since $\hat{x}_y$ is the minimum point of the $C^2$ function $\Phi(\cdot, y)$, then the Hessian of $\Phi(\cdot, y)$ at $\hat{x}_y$ is positive definite,

$$\partial_{xx}^2 \Phi(\hat{x}_y, y) = \nabla_{xx}^2 (\varphi(x) + c(x, y))\Big|_{x=\hat{x}_y} = \nabla^2 \varphi(\hat{x}_y) + \partial_{xx}^2 c(\hat{x}_y, y) \succ 0.$$

Since $\partial_{xx}^2 \Phi(\hat{x}_y, y)$ is positive definite, it is also invertible. We can now apply the implicit function theorem to show that the equation $\partial_x \Phi(x, y) = 0$ determines an implicit function $\hat{x}(\cdot)$, which satisfies $\hat{x}(y) = \hat{x}_y$ in a small neighbourhood $U \subset \mathbb{R}^n$ containing $y$. Furthermore, one can show that $\hat{x}(\cdot)$ is continuously differentiable at $y$. We will denote $\hat{x}_y$ as $\hat{x}(y)$ in our following discussion.

Now differentiating equation 12 with respect to $y$ yields

$$\partial_{xx}^2 \Phi(\hat{x}(y), y) \nabla \hat{x}(y) + \partial_{xy}^2 c(\hat{x}(y), y) = 0 \tag{13}$$

On one hand, equation 13 tells us

$$\nabla \hat{x}(y) = -\partial_{xx} \Phi(\hat{x}(y), y)^{-1} \partial_{xy} c(\hat{x}(y), y). \tag{14}$$

On the other hand, notice that $c \in C^2(\mathbb{R}^n \times \mathbb{R}^n)$, thus $\partial_{xy} c = \partial_{yx} c$. By equation 13, we have

$$\begin{aligned} \partial_{yx}^2 c(\hat{x}(y), y) \nabla \hat{x}(y) &= -\partial_{xx}^2 \Phi(\hat{x}(y), y) \nabla \hat{x}(y) \nabla \hat{x}(y) \\ &= -(\nabla^2 \varphi(\hat{x}(y)) + \partial_{xx}^2 c(\hat{x}(y), y)) \nabla \hat{x}(y) \nabla \hat{x}(y). \end{aligned} \tag{15}$$

Now we are able to prove our theorem, we directly compute

$$\nabla^2 \Psi_x(y) = \nabla^2 f(y) - \partial_{yy}^2 c(x, y). \tag{16}$$

in order to compute $\nabla^2 f(y)$, we first compute $\nabla f(y)$

$$\nabla f(y) = \nabla(\varphi(\hat{x}(y)) + c(\hat{x}(y), y)) = \partial_y c(\hat{x}(y), y). \tag{17}$$

the second equality is due to the envelope theorem Afriat (1971). Then $\nabla^2 f(y)$ can be computed as

$$\nabla^2 f(y) = \partial_{yx} c(\hat{x}(y), y) \nabla \hat{x}(y) + \partial_{yy} c(\hat{x}(y), y). \tag{18}$$

Plugging equation 15 into equation 18, recall equation 16, this yields

$$\nabla^2 \Psi_x(y) = -(\nabla^2 \varphi(\hat{x}(y)) + \partial_{xx}^2 c(\hat{x}(y), y)) \nabla \hat{x}(y) \nabla \hat{x}(y) + \partial_{yy}^2 c(\hat{x}(y), y) - \partial_{yy}^2 c(x, y)$$

Now due to the assumption that $\partial_{yy} c(x, y)$ is independent of $x$, one has $\partial_{yy}^2 c(\hat{x}(y), y) - \partial_{yy}^2 c(x, y) = 0$. As a result we obtain

$$\begin{aligned} \nabla^2 \Psi_x(y) &= -(\nabla^2 \varphi(\hat{x}(y)) + \partial_{xx}^2 c(\hat{x}(y), y)) \nabla \hat{x}(y) \nabla \hat{x}(y) \\ &= -\partial_{xx}^2 \Phi(\hat{x}(y), y) \nabla \hat{x}(y) \nabla \hat{x}(y). \end{aligned} \tag{19}$$

To further simplify equation 19, recall equation 14, we have

$$\nabla^2 \Psi_x(y) = -\partial_{xy} c(\hat{x}(y), y) \partial_{xx} \Phi(\hat{x}(y), y)^{-1} \partial_{xy} c(\hat{x}(y), y).$$

By equation 11, $\lambda(y) I_n \succeq \partial_{xx} \Phi(\hat{x}(y), y) \succ O_n$. Recall the assumption that $\partial_{xy} c$ is self-adjoint, and equation 10 leads to $\sigma_{\min}(\partial_{xy} c(x, y)) = \sigma(x, y)$. Now applying lemma 1 yields

$$\nabla^2 \Psi_x(y) \preceq -\frac{\sigma(x, y)^2}{\lambda(y)} I_n.$$

$\square$

Now we can prove main result in Theorem 2:

*Proof of Theorem equation 2.* In this proof, we denote $\int$ as $\int_{\mathbb{R}^d}$ for simplicity.

We first recall

$$\mathcal{L}(T, f) = \int f(y) \rho_b(y) \, dy - \int (f(T(x)) - c(x, T(x))) \rho_a(x) dx,$$

also recall definition equation 7, $f^{c,-}(x) = \sup_y\{f(y) - c(x,y)\}$, we can write

$$\mathcal{E}_1(T, f) = -\int[f(T(x)) - c(x, T(x))]\rho_a \, dx + \inf_{\widetilde{T}}\left\{\int[f(\widetilde{T}(x)) - c(x, \widetilde{T}(x))]\rho_a \, dx\right\}$$

$$= \int[f^{c,-}(x) - (f(T(x)) - c(x, T(x)))]\rho_a(x)dx$$

We denote

$$T_f(x) = \operatorname{argmax}_y\{f(y) - c(x, y)\} = \operatorname{argmax}_y\{\Psi_x(y)\},$$

then we have

$$\nabla\Psi_x(T_f(x)) = 0. \tag{20}$$

On the other hand, one can write:

$$\mathcal{E}_1(T, f) = \int[(f(T_f(x)) - c(x, T_f(x))) - (f(T(x)) - c(x, T(x)))]$$

$$= \int[\Psi_x(T_f(x)) - \Psi_x(T(x))]\rho_a(x) \, dx$$

For a fixed $x$, since $\Psi_x(\cdot) \in C^2(\mathbb{R}^n)$, then

$$\Psi_x(T(x)) - \Psi_x(T_f(x)) = \nabla\Psi_x(T_f(x))(T(x) - T_f(x)) + \frac{1}{2}(T(x) - T_f(x))^{\mathrm{T}}\nabla^2\Psi_x(\omega(x))(T(x) - T_f(x))$$

with $\omega(x) = (1 - \theta_x)T(x) + \theta_x T_f(x)$ for certain $\theta_x \in (0, 1)$. By equation 20 and Lemma 2, we have

$$\Psi_x(T(x)) - \Psi_x(T_f(x)) \leq -\frac{\sigma(x, \omega(x))^2}{2\lambda(\omega(x))}|T(x) - T_f(x)|^2.$$

Thus we have:

$$\mathcal{E}_1(T, f) = \int[\Psi_x(T_f(x)) - \Psi_x(T(x))]\rho_a(x) \, dx \geq \int\frac{\sigma(x, \omega(x))^2}{2\lambda(\omega(x))}|T(x) - T_f(x)|^2\rho_a(x) \, dx \tag{21}$$

On the other hand, let us denote the optimal Monge map from $\rho_a$ to $\rho_b$ as $T_*$, by Kontorovich duality, we have

$$\sup_f\inf_T \mathcal{L}(T, f) = \inf_{T, T_\sharp\rho_a = \rho_b}\int c(x, T(x))\rho_a \, dx = \int c(x, T_*(x))\rho_a \, dx$$

Thus we have

$$\mathcal{E}_2(f) = \int c(x, T_*(x))\rho_a \, dx - \left(\int f(y)\rho_b \, dy - \int f^{c,-}(x)\rho_a \, dx\right)$$

$$= \int c(x, T_*(x))\rho_a \, dx - \left(\int f(T_*(x))\rho_a \, dx - \int f^{c,-}(x)\rho_a \, dx\right)$$

$$= \int[f^{c,-}(x) - (f(T_*(x)) - c(x, T_*(x)))]\rho_a \, dx$$

Similar to the previous treatment, we have

$$\mathcal{E}_2(f) = \int[\Psi_x(T_f(x)) - \Psi_x(T_*(x))]\rho_a(x) \, dx$$

Apply similar analysis as before, we will also have

$$\mathcal{E}_2(f) \geq \int\frac{\sigma(x, \xi(x))^2}{2\lambda(\xi(x))}|T_*(x) - T_f(x)|^2\rho_a(x) \, dx \tag{22}$$

with $\xi(x) = (1 - \tau_x)T_*(x) + \tau_x T_f(x)$ for certain $\tau_x \in (0, 1)$.

Now we set

$$\beta(x) = \min\left\{\frac{\sigma(x, \omega(x))}{2\lambda(\omega(x))}, \frac{\sigma(x, \xi(x))}{2\lambda(\xi(x))}\right\}, \tag{23}$$

combining equation 21 and equation 22, we obtain

$$\mathcal{E}_1(T, f) + \mathcal{E}_2(f) \geq \int \beta(x)(|T(x) - T_f(x)|^2 + |T_*(x) - T_f(x)|^2)\rho_a \, dx$$

$$\geq \int \frac{\beta(x)}{2}|T(x) - T_*(x)|^2 \rho_a \, dx$$

This leads to $\|T - T_*\|_{L^2(\beta\rho_a)} \leq \sqrt{2(\mathcal{E}_1(T, f) + \mathcal{E}_2(f))}$.

$\square$

## D  ADDITIONAL RESULTS ON SYNTHETIC DATASETS

**Learning with unequal dimensions**  Our algorithm framework enjoys a distinguishing quality that it can learn the map from a lower dimension space $\mathbb{R}^n$ to a manifold in a higher dimension space $\mathbb{R}^m (n \leq m)$. In this scenario, we make the input dimension of neural network $T$ to be $n$ and output dimension to be $m$. In case the cost function $c(x, y)$ requires dimensions are $x$ and $y$ are equal dimensional, we patch zeros behind each sample $X \sim \rho_a$ and complement to a counterpart sample $\widetilde{X} = [X; \mathbf{0}]$, where dimension of $\mathbf{0}$ is $m - n$. And the targeted min-max problem is replaced by

$$\max_{\theta} \min_{\eta} \frac{1}{N} \sum_{k=1}^{N} c(\widetilde{X}_k, T_\theta(X_k)) - f_\eta(T_\theta(X_k)) + f_\eta(Y_k).$$

In Figure 3, we conduct one experiment for $n = 1$ and $m = 2$. The incomplete ellipse is a 1D manifold and our algorithm is able to learn a symmetric map from $\mathcal{N}(0, 1)$ towards it.

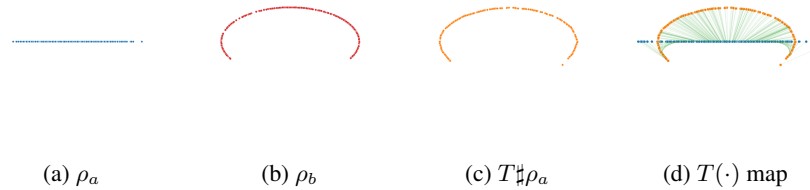

(a) $\rho_a$          (b) $\rho_b$          (c) $T\sharp\rho_a$          (d) $T(\cdot)$ map

Figure 3: Qualitative results for learning unequal dimension maps. $\rho_a$ for two examples are both $\mathcal{N}(0, 1)$, and $\rho_b$ are uniformly distributed on a incomplete ellipse and a ball respectively.

**Decreasing function as the cost**  We consider the cost function $c(x, y) = \phi(|x - y|)$ with $\phi$ as a monotonic decreasing function. We test our algorithm for a specific example $\phi(s) = \frac{1}{s^2}$. In this example, we compute for the optimal Monge map from $\rho_a$ to $\rho_b$ with $\rho_a$ as a uniform distribution on $\Omega_a$ and $\rho_b$ as a uniform distribution on $\Omega_b$, where we define

$$\Omega_a = \{(x_1, x_2) \mid 6^2 \geq x_1^2 + x_2^2 \geq 4^2\}, \quad \Omega_b = \{(x, x_2) \mid 2^2 \geq x_1^2 + x_2^2 \geq 1^2\}.$$

We also compute the same problem for $L^2$ cost. Figure 4 shows the transported samples as well as the differences between two cost functions.

**Monge problem on sphere**  For a given sphere $S$ with radius $R$, for any two points $x, y \in S$, we define the distance $d(x, y)$ as the length of the geodesic joining $x$ and $y$. Now for given $\rho_a, \rho_b$ defined on $S$, we consider solving the following Monge problem on $S$

$$\min_{T, \ T_\sharp\rho_a = \rho_b} \int_S d(x, T(x))\rho_a(x) \, dx. \tag{24}$$

Such sphere OT problem can be transferred to an OT problem defined on angular domain $D = [0, 2\pi) \times [0, \pi]$, to be more specific, we consider $(\theta, \phi)$ ($\theta \in [0, 2\pi)$, $\phi \in [0, \pi]$) as the azimuthal and polar angle of the spherical coordinates. For two points $x = (R \sin\phi_1 \cos\theta_1, R \sin\phi_1 \sin\theta_1, R \cos\phi_1)$, $y = (R \sin\phi_2 \cos\theta_2, R \sin\phi_2 \sin\theta_2, R \cos\phi_2)$ on $S$, the geodesic distance

$$d(x, y) = c((\theta_1, \phi_1), (\theta_2, \phi_2)) = R \cdot \arccos(\sin\phi_1 \sin\phi_2 \cos(\theta_2 - \theta_1) + \cos\phi_1 \cos\phi_2).$$

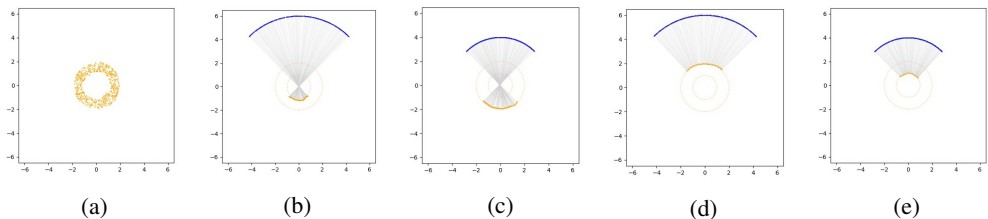

Figure 4: (a) samples of computed $T_\sharp \rho_a$; $c(x,y) = \frac{1}{|x-y|^2}$: Computed Monge map of quarter circles with radius $6$ (subplot b) and radius $4$ (subplot c); $c(x,y) = |x-y|^2$: Computed Monge map of quarter circles with radius $6$ (subplot d) and radius $4$ (subplot e).

Denote the corresponding distribution of $\rho_a, \rho_b$ on $D$ as $\hat{\rho}_a, \hat{\rho}_b$, now equation 24 can also be formulated as

$$\min_{\hat{T}, \hat{T}_\sharp \hat{\rho}_a = \hat{\rho}_b} \int c((\theta, \phi), \hat{T}(\theta, \phi)) \hat{\rho}_a \, d\theta d\phi. \tag{25}$$

We set $\hat{\rho}_a = U([0, 2\pi]) \otimes U([0, \frac{\pi}{4}])$ and $\hat{\rho}_b = U([0, 2\pi]) \otimes U([\frac{3\pi}{4}, \pi])$. We apply our algorithm to solve equation 25 and then translate our computed Monge map back to the sphere $S$ to obtain the following results

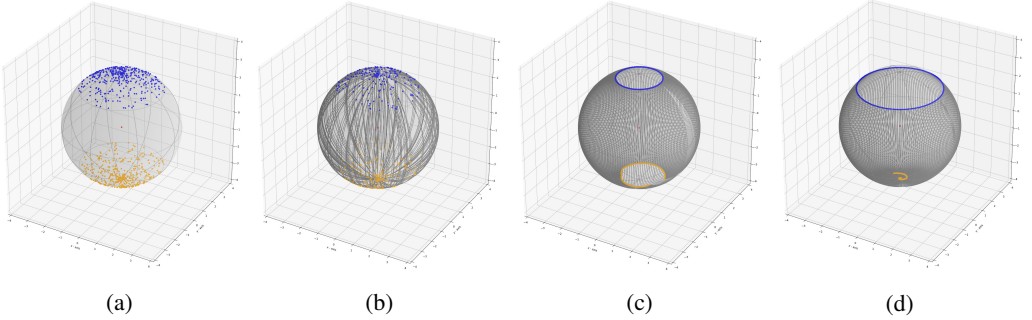

Figure 5: Monge map from $\rho_a$ to $\rho_b$ on the sphere: (a) blue samples from $\rho_a$ (corresponds to $\hat{\rho}_a$) and orange samples from $\rho_b$ (corresponds to $\hat{\rho}_b$); (b) blue samples from $\rho_a$, orange samples are obtained from $\hat{T}_\sharp \hat{\rho}_b$, grey curves are geodesics connecting each transporting pairs; (c) our computed Monge map maps blue ring ($\phi = \frac{\pi}{8}$) to the orange curve (ground truth is $\phi = \frac{7}{8}\pi$); (d) our computed Monge map maps blue ring ($\phi = \frac{\pi}{4}$) to the orange curve (ground truth is the southpole)

### D.1 QUANTITATIVE RESULTS

In unpaired inpainting task, we also evaluate Fréchet Inception Distance (Heusel et al., 2017) of the generated composite images w.r.t. the original images on the test dataset. We use $40k$ images in total and compute the score with the implementation provided by Obukhov et al. (2020). The results are presented in Table 1. It shows that the transportation cost $c(x, y)$ substantially promotes a map that generates more realistic images.

Table 1: Quantitative evaluation results on CelebA $64 \times 64$ test dataset.

|  | $\alpha = 0$ | $\alpha = 10$ | $\alpha = 10000$ |
|---|---|---|---|
| FID | 18.7942 | 9.2857 | **3.7109** |

# E HYPER-PARAMETERS

## E.1 SYNTHETIC DATASETS

**Unequal dimensions** The networks $T_\theta$ and $f_\eta$ each has 5 layers with 10 hidden neurons. The batch size $B = 100$. $K_1 = 6, K_2 = 1$. The learning rate is $10^{-3}$. The number of iterations $K = 12000$.

**Decreasing cost function** In this example, we set $T_\theta(x) = x + F_\theta(x)$ and optimize over $\theta$. For either $\frac{1}{|x-y|^2}$ or $|x-y|^2$ case we set both $F_\theta$ and the Lagrange multiplier $f_\eta$ as six layers fully connected neural networks, with PReLU and Tanh activation functions respectively, each layer has 36 nodes. The training batch size $B = 2000$. We set $K = 2000, K_1 = 8, K_2 = 6$.

**On sphere** In this example, we set $T_\theta(x) = x + F_\theta(x)$ and optimize over $\theta$. We set both $F_\theta$ and $f_\eta$ as six layers MLP, with PReLU activation functions, each layer has 8 nodes The training batch size $M = 200$. We set $K = 4000, K_1 = 8, K_2 = 4$. We choose rather small learning rate in this example to avoid gradient blow up, we set $0.5 \times 10^{-5}$ as the learning rate for $\theta$ and $10^{-5}$ as the learning rate for $\eta$.

## E.2 UNPAIRED INPAINTING

The loss function is slightly different with the equation 3. We modify the $f(T(x))$ to be $f(G(x))$ to strengthen the training of $f$

$$\sup_f \inf_T \int_{\mathbb{R}^n} \left[ c(x, T(x)) - f(G(x)) \right] \rho_a(x) \, dx + \int_{\mathbb{R}^m} f(y)\rho_b(y) \, dy.$$

In the unpaired inpainting experiments, the images are first cropped at the center with size 140 and then resized to $64 \times 64$ or $128 \times 128$. We choose learning rate to be $1 \cdot 10^{-3}$, Adam (Kingma & Ba, 2014) optimizer with default beta parameters, $K_2 = 1$. The batch size is 64 for CelebA64 and 16 for CelebA128. The number of inner loop iteration $K_1 = 5$ for CelebA64 and $K_1 = 10$ for CelebA128.

We use exactly the same UNet for the map $T$ and convolutional neural network for $f$ as Rout et al. (2022, Table 9) for CelebA64 and add one additional convolutional block in $f$ network for CelebA128.

On NVIDIA RTX A6000 (48GB), the training time of CelebA64 experiment is 10 hours and the time of CelebA128 is 45 hours.

