# OpenReview forum: "Scalable Computation of Monge Maps with General Costs"
_ICLR.cc/2022/Workshop/DGM4HSD — ICLR 2022 DGM4HSD workshop Poster_

### Official Review · Reviewer_s4yq · 2022-03-25
**A clearly written paper introduced an efficient algorithm to compute the Morgan map between two potential dimensional distributions in different dimensions. The paper is supported by results from both synthetic and real data and theoretical analysis and proofs.**

**Rating:** 7
**Confidence:** 4

**Review:**

The paper introduced an efficient algorithm to compute the Morgan map between two potentially high-dimensional distributions in different dimensions.

The paper is well written, with details proofs presented in supplementals. The pseudocode is given in the main text for readers to implement the algorithm. Some preliminary results on unpaired image inpainting are presented in the main text, and hyperparameter setting is given in the supplemental. There are several demos on synthetic data presented in supplementals for readers to better understand their approach.

The authors demonstrate the effectiveness of their approach on an unpaired inpainting problem using the CelebA dataset. Empirical results show the algorithm work but qualitatively. The authors also demonstrate the influence of the hyperparacter alpha on the inpainting results. The Fréchet Inception Distances are presented in supplementals. Unfortunately, there are no quantitative comparison results with other methods, no reporting of training time etc.

The paper is only marginally related to the theme of the workshop: Deep Generative Models for Highly Structured Data, but I think the method could be of potential interest to the audience.

The paper is well written and easy to follow. Although it seems that the code is not available, all hyperparameters are presented in supplementals for interested readers to reproduce the results.

---

### Official Review · Reviewer_BgxT · 2022-03-28
**A min-max approach to generalized optimal transport**

**Rating:** 5
**Confidence:** 2

**Review:**

This paper proposes a min-max optimization problem to find solutions to the primal Monge problem.

Positives:
- While the approach of solving the optimal transport problems by using their dual is fairly standard, I think theorem 2 is interesting ( subject to my questions below).
- The experiments show that this approach can work with unpaired inpainting samples. While it is possible to visually pair the original and masked images, the algorithm does not use this knowledge, and I think it's interesting.

Negatives:
- The paper proposes a min-max optimization problem to find the Monge solution. However, the complexity of solving this min-max problem is never considered.
- Some of the assumptions are placed in the theorem statement, such as in theorem 2. Please move them to the assumptions para that precedes the theorem.
- Theorem 2 does not make sense to me. Can't the weighting function $\beta$ be arbitrarily small such that the upper bound is achieved?
- The naming convention is quite problematic. The true Monge problem is ill-defined as its feasible set is not compact (for example, how would you find a transport map from the discrete distribution on {0,1} to the continuous distribution on [0,1] ?). As this paper is considering general problems where $\rho_a$ and $\rho_b$ need not be defined on the same probability space (for e.g., R^n and R^m), the authors are actually using the Kantorovich relaxation.

Minor:

- Below eqn 1, the definition of the push-forward is incorrect. It should be $ T \rho_a ( E ) = \rho_a ( T^{-1} (E) )$ , not $ \rho_b ( T^{-1} (E) )$ as it is currently defined.

- Some of the claims are unsubstantiated. For example, the authors say that they do not need absolute continuity of the marginals, but this is never shown.

---

### Decision · Program_Chairs · 2022-03-26

Accept (Poster)